# Orthopedic and Nonsurgical Orthodontic Treatment of Adolescent Skeletal Class III Malocclusion Using Bone-Anchored Maxillary Protraction and Temporary Anchorage Devices: A Case Report

**DOI:** 10.3390/children9050683

**Published:** 2022-05-07

**Authors:** Mohammed Alnefaie, Woo-Jin Han, Yoon-Soo Ahn, Won-Kyeong Baik, Sung-Hwan Choi

**Affiliations:** Department of Orthodontics, Institute of Craniofacial Deformity, Yonsei University College of Dentistry, Seoul 03722, Korea; dr-moh99@hotmail.com (M.A.); woowaa30@naver.com (W.-J.H.); ays0608@naver.com (Y.-S.A.); wkbaik1@naver.com (W.-K.B.)

**Keywords:** skeletal Class III, bone-anchored maxillary protraction, miniplate, protraction, intermaxillary elastics

## Abstract

This report describes the case of a 12-year-old female patient with a long mandible experiencing difficulty chewing with the right molar. Considering the age of the patient, bone-anchored maxillary protraction using four miniplates placed below the maxillary zygomatic arch and anterior symphysis of the mandible and Class III intermaxillary elastics were planned. After 12 months, orthodontic treatment was initiated. After extraction of the impacted maxillary right second premolar and mandibular right second primary molar, protraction of the mandibular right molars was performed using a miniplate placed on the anterior part of the mandible as an anchor. Miniscrews were placed in the left posterior part of the mandible to improve the molar relationship and correct the dental midline through distalization of the mandibular left posterior teeth. We reported successful sequential comprehensive nonsurgical treatment in an adolescent with skeletal Class III malocclusion.

## 1. Introduction

Orthodontic management of young patients with skeletal Class III malocclusion is challenging because most skeletal Class III patients and their guardians do not prefer orthognathic surgery as a treatment plan after growth completion [1].

The conventional standard for early skeletal Class III treatment involves the use of a facemask. However, because the facemask uses teeth as the anchor source and applies a direct force to the teeth, it causes unwanted tooth movement or shows low efficiency in skeletal changes [2]. When using the facemask, the anterior traction of the maxilla is limited, and clockwise rotation of the mandible, labial inclination of the maxillary incisors, and lingual inclination of the mandibular incisors are known as the main effects [3,4]. Recently, as temporary anchorage devices such as miniscrews and miniplates are commonly used, attempts to reduce the dental effect of the existing facemask and increase the skeletal effect have been reported [1,5].

Kokich et al. [6] reported the merits of maxillary traction with a facemask using the intentionally ankylosed deciduous canines as anchorage sources. Smalley et al. [7] reported significant maxillary traction results by experimenting with a Branemark-style implant as anchorage sources in monkeys (*Macaca nemestrina*). Recently, the most effective approach to maxillary protraction has been demonstrated to be Class III intermaxillary elastics between bone-anchored maxillary protraction with miniplates at the base of the zygoma above the maxillary molars and the anterolateral surface of the mandible, such that light forces are delivered to the jaw rather than the teeth [8]. The advantage of this approach is that greater skeletal change is possible than with conventional facemask treatment, and traction is possible throughout the day without the need for patient cooperation [9]. Bone-anchored maxillary protraction can be applied to patients above 11 years of age.

Here, we report the case of a 12-year-old female patient with skeletal Class III malocclusion, in whom a good skeletal and dental relationship was obtained through nonsurgical orthodontic treatment using bone-anchored maxillary protraction.

## 2. Case Presentation

### 2.1. Diagnosis and Etiology

A 12-year-old girl visited the orthodontic department at Yonsei University Dental Hospital in Seoul, South Korea. She had previously been treated with a Frankel regulator III (FR III) functional appliance for mandibular protrusion at a private clinic when she was 7 years old. The clinical presentation was as follows:-Open bite in the right premolar area because of the failure of eruption of the maxillary right second premolar with prolonged retention of the maxillary and mandibular primary second molars (Figure 1).-Mandibular protrusion.

During clinical examination of the patient, a straight face was observed on the frontal extraoral photograph (Figure 1). Intraoral examination showed a crossbite in the maxillary right lateral incisor, and the overjet and overbite were both 1.0 mm with no functional shift (Figure 1). The midline of the maxillary and mandibular dentitions deviated by 1 mm to the left and right, respectively, compared to the facial midline. During smiling, a lateral open bite in the right premolar area was identified. Retained maxillary and mandibular primary second molars were also noted. The maxillary and mandibular arch length discrepancies were 1.5 and 0.5 mm, respectively.

Lateral cephalometric analysis showed an SNA (angle formed by the lines connecting the sella, nasion, and point A) of 70.9°, an SNB angle of 72.7°, and an ANB angle of −1.9° (Table 1 and Figure 2A). Both the maxillary and mandibular incisors were lingually inclined, while the upper lip was retruded according to Ricketts’ esthetic line. The cervical vertebrae maturation index (CVMI) indicated that the patient was between stages 5 and 6 and had surpassed the maximum growth period. Panoramic radiography showed that the maxillary right second premolar was inversely impacted (Figure 2B) and the mandibular right second premolar was congenitally missing, with retained maxillary and mandibular primary second molars. Cone beam computed tomography (CBCT) was performed to determine the exact location of the maxillary right second premolar (Figure 2C,D). As shown in Figure 2C, it was clearly confirmed that the radiopacity of the maxillary sinus around the right impacted tooth was increased differently from that of the left maxillary sinus. Therefore, the reverse direction of the premolar towards the maxillary sinus caused perforation of the sinus wall, resulting in right sinusitis.

### 2.2. Treatment Objectives

Based on the clinical and radiographic findings, the patient was diagnosed with skeletal Class III malocclusion. This diagnosis included an anterior crossbite, sinusitis on the right side (suspected), ectopic impaction state (inverted) of the maxillary right second premolar, and congenitally missing mandibular premolar with prolonged retention of the mandibular right primary second molar. The following treatment objectives were planned: (1) treatment of maxillary sinusitis, (2) improvement of the skeletal pattern and anterior crossbite, (3) crowding relief and restorative treatment for impacted and congenitally missing teeth in the maxilla and mandible, (4) dental midline correction, and (5) improvement of the soft tissue profile.

### 2.3. Treatment Alternatives

First, right maxillary sinusitis was caused by the inverted impacted tooth that penetrated the sinus wall. Although maxillary sinusitis cannot be treated directly at the orthodontic department, it was confirmed that the impacted maxillary right second premolar was the cause of right maxillary sinusitis, and that the maxillary sinusitis could be improved if the cause was removed. Moreover, the prognosis would be poor if orthodontic traction of the tooth was performed. Therefore, tooth extraction was selected as the best treatment option.

The main cause of the patient’s skeletal Class III relationship was the deficient growth of the maxilla, in contrast to the normal mandibular growth. Based on the treatment objectives, the following alternatives were considered:

(1) Orthopedic treatment with maxillary protraction using a conventional facemask or bone-anchored maxillary traction with Class III intermaxillary elastics. However, CVMI revealed that the patient was between stages 5 and 6. Therefore, the maximum growth period had already been surpassed, whereas the ideal time for using a conventional facemask is during CVMI stage 1 [10]. Therefore, the rate of clinical success in resolving the skeletal discrepancy would be greater with bone-anchored maxillary traction with Class III intermaxillary elastics than with a facemask.

(2) Orthognathic surgery after the completion of skeletal growth.

As the patient did not want surgical treatment, option 1, using bone-anchored maxillary traction with Class III intermaxillary elastics, was chosen.

## 3. Treatment Progress

First, we referred the patient to the oral and maxillofacial surgery department for the extraction of the impacted tooth and placement of the miniplate. The miniplates were placed under general anesthesia. The maxillary plate was placed below the zygoma so that the hook was located between the second premolar and first molar (Figure 3A). In addition, the miniplate was placed in the symphysis area so that the hook was located between the lateral incisor and the canine. A wafer device was worn by the patient for 1 week after surgical extraction of the impacted tooth. For 11 months, the patient wore Class III intermaxillary elastics daily (Figure 3B).

In the second phase, after bonding the fixed appliance to the maxillary and mandibular dentition, treatment started with the following goals: proclining maxillary incisors and closing most of the extraction space of the extracted mandibular primary tooth through protraction of the right molar, which induced spontaneous eruption of the right third molar (Figure 4). In addition, to improve the midline, the mandibular left third molar was extracted, and the mandibular dentition was moved to the left.

An anterior miniplate was used for protraction of the mandibular posterior molars after the extraction of the right primary second molar (Figure 5A). Since the midline of the maxillary dentition was also deviated to the left, total arch distalization was performed towards the right side of the maxillary dentition using the miniplate on the right side of the maxilla to improve the midline. A miniscrew was placed on the left side of the mandible, and distal movement of the dentition was performed to correct the midline and improve the occlusal relationship in the posterior region.

After 39 months of secondary orthodontic treatment, all spaces were closed, the anterior crossbite and lateral open bite improved, and stable occlusion was achieved (Figure 6). After treatment, most cephalometric values, except SNA, improved within the normal range (Figure 7A, Table 1). Panoramic radiography demonstrated that the mandibular right third molar erupted in an upright position (Figure 7B). No significant root resorption was observed in the maxillary anterior teeth, but the periodontal ligament was clearly widened. It was observed that the right maxillary sinus pneumatization was severe up to the place where the impacted tooth was in the past.

## 4. Discussion

The patient was previously treated with an FR III functional device for mandibular protrusion at a private clinic when she was 7 years old. Having observed that no surgery would be needed in the future, we decided to urgently resolve her skeletal discrepancy. The treatment goals were achieved using miniplates and intermaxillary elastics known as bone-anchored maxillary protraction, followed by fixed orthodontic therapy.

The primary second molars were present on the right side of the maxilla and mandible. Thus, in the second phase of treatment, it was important to decide whether to maintain or extract the teeth. The primary mandibular second molar was covered by a metal crown with severe root resorption, which may be considered a poor prognosis. Additionally, straight eruption of the third molar could be expected if the primary molar in the mandible was extracted and a good occlusal relationship could be established. There was almost no mobility of the maxillary primary tooth at the start of the second phase. Moreover, severe right maxillary sinus congestion was also noted (sinus pneumatization) because of the impacted second premolar. Therefore, protraction of the posterior teeth in this area would have been associated with various side effects, such as root resorption and perforation of the maxillary sinus wall during prolonged treatment. Therefore, only the mandibular primary tooth was extracted while maintaining the maxillary primary tooth.

When superimposition analysis was performed with pre- and post-treatment radiographs, a small amount of growth was observed in the mandible (Figure 8). Protraction of the mandibular right molars and retraction of the mandibular anterior teeth were successful after extraction of the mandibular primary second molars.

The limitations of this case are as follows. Fortunately, no adverse events such as temporomandibular disorder or gingival recession in the mandibular anterior region occurred before and after the use of bone-anchored maxillary protraction in this patient. However, the treatment protocol that can prevent the possibility of such side effects was also performed without being prepared; so, it is necessary to secure a response for this in the future. Fortunately, no significant root resorption was observed while attempting the labial inclination of the maxillary anterior teeth in the second phase, but the periodontal ligament was clearly widened, and the labial inclination was also caused by the unintentional and uncontrolled tipping movement. Therefore, to clearly confirm whether the treatment result of this case is stably maintained, it is thought that at least 1–2 years of maintenance period observation is necessary.

## 5. Conclusions

Class III malocclusion is a challenging anomaly. Furthermore, most patients with Class III malocclusion do not always prefer surgical correction. Thus, it is important to pay more attention to patient assessment and selection during the diagnosis and treatment planning.

The present case shows successful results of orthopedic and orthodontic treatment in an adolescent girl who exhibited characteristics of Class III malocclusion. Effective management and treatment of Class III malocclusion can be achieved in adolescents through nonsurgical orthodontic treatment. The skeletal and dental relationships were improved, thereby enhancing the appearance of the mid-face, lip profile, and chin position.

## Figures and Tables

**Figure 1 children-09-00683-f001:**
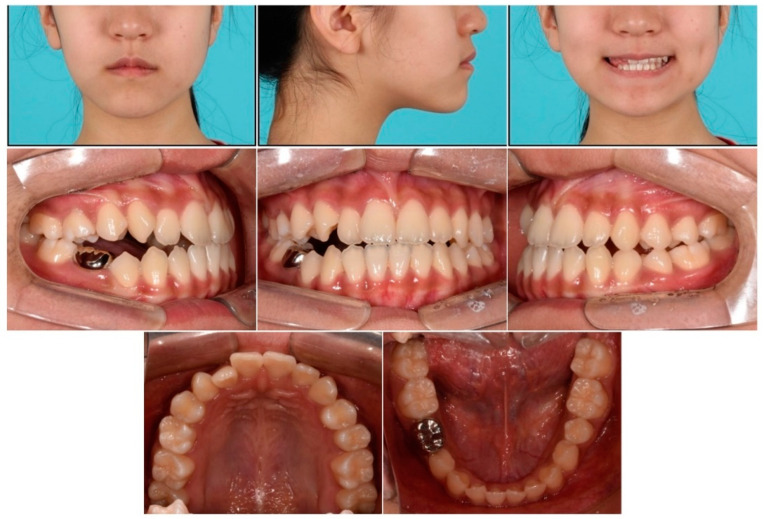
Pretreatment facial and intraoral photographs.

**Figure 2 children-09-00683-f002:**
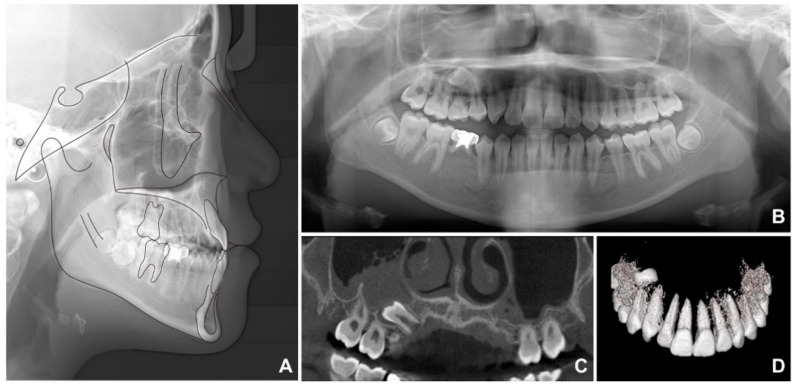
Pretreatment radiographs: (**A**) lateral cephalogram; (**B**) panoramic radiograph; (**C**,**D**) cone beam computed tomography.

**Figure 3 children-09-00683-f003:**
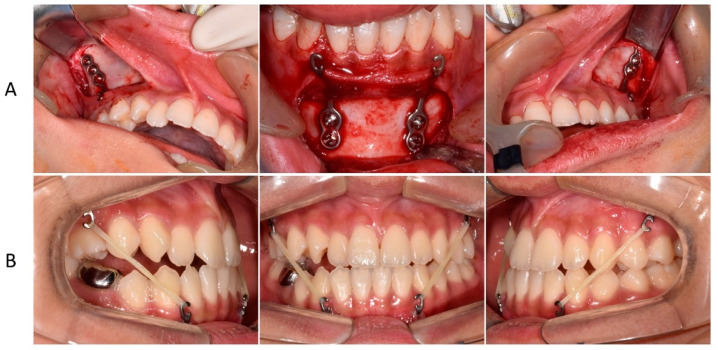
Insertion and use of the miniplate: (**A**) surgery for miniplate insertion; (**B**) use of Class III inter-maxillary elastics.

**Figure 4 children-09-00683-f004:**
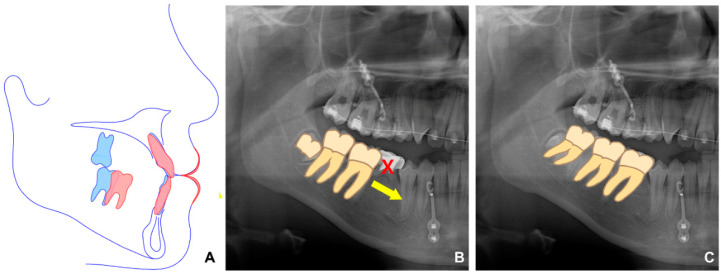
Schematic diagram related to protraction of the mandibular right posterior teeth. (**A**) Virtual treatment objective image. Blue and red lines are before and after treatment, respectively. (**B**) The protraction of the mandibular right first and second molars to the position where the right primary second molar was extracted. The yellow arrow represents the direction of tooth movement. X indicates extracted tooth. (**C**) Eruption of the mandibular third molar to the natural occlusal plane because of the anterior movement of the mandibular right molars.

**Figure 5 children-09-00683-f005:**
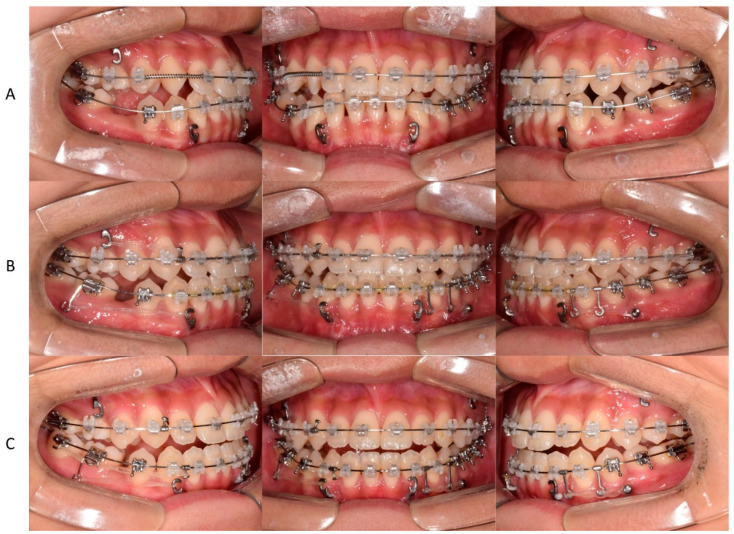
Nonsurgical orthodontic treatment process using a fixed appliance. (**A**) space regaining for maxillary right lateral incisor arrangement after bonding the fixed appliance. (**B**) Closing the mandibular extraction space using the miniplate anchorage, moving the maxillary midline to the right, and moving the mandibular midline to the left. (**C**) The mandibular extraction space was closed.

**Figure 6 children-09-00683-f006:**
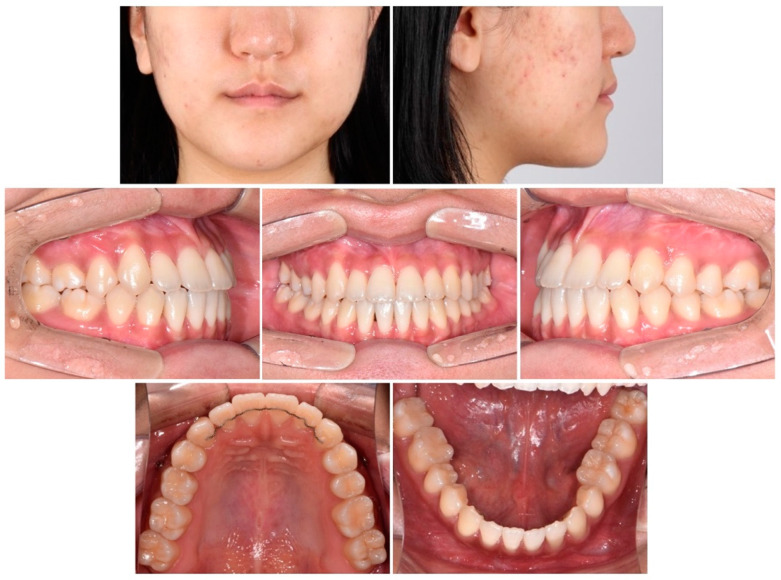
Intraoral and extraoral photographs after debonding.

**Figure 7 children-09-00683-f007:**
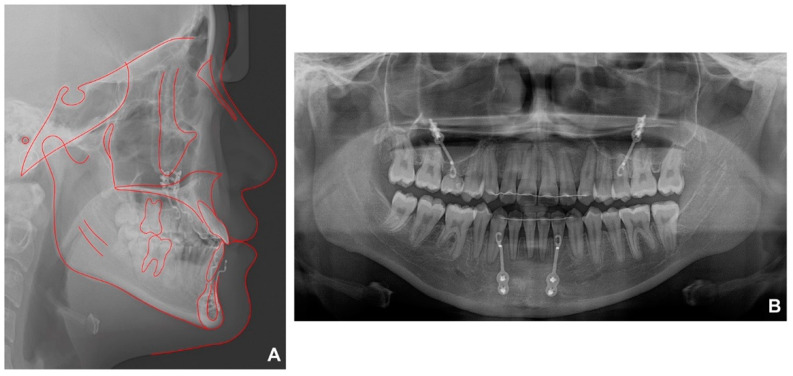
Radiographs after debonding: (**A**) lateral cephalometric radiographs; (**B**) panoramic radiograph.

**Figure 8 children-09-00683-f008:**
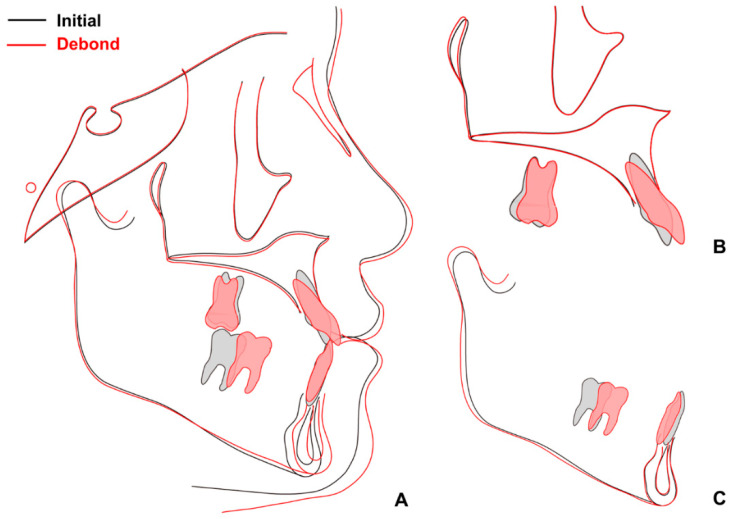
Superimposition before and after treatment: (**A**) total; (**B**) maxilla; (**C**) mandible.

**Table 1 children-09-00683-t001:** Lateral cephalometric measurements.

Measurement	Normal Value	Pre-Treatment (12 y 11 m)	Phase I Treatment (13 y 9 m)	Phase II Treatment (17 y 7 m)
**Skeletal**				
SNA (°)	81.9 ± 3.0	70.9	73.2	75.6
SNB (°)	78.0 ± 3.0	72.7	73.6	76.4
ANB (°)	4.0 ± 2.0	−1.9	−0.4	−0.8
Wits (mm)	−2.0 ± 2.4	−11.2	−9.0	−4.8
SN-GoMe (°)	36.0 ± 4.0	42.0	40.9	36.7
Gonial angle (°)	122.0 ± 6.0	122.5	120.0	118.3
**Dental**				
U1 to SN (°)	105.0 ± 5.0	96.3	100.5	110.8
L1 to GoMe (°)	95.0 ± 4.0	78.7	76.7	84.7
**Soft tissue**				
Nasolabial angle (°)	94.4 ± 10.3	88.3	92.6	91.5
Upper lip to E line (mm)	1.0 ± 2.0	−2.6	−4.2	−3.3
Lower lip to E line (mm)	2.0 ± 2.0	0.2	−1.2	−1.4

SNA, angle consisting of sella, nasion, and point A; SNB, angle consisting of sella, nasion, and point B; ANB, angle consisting of point A, nasion, and point B; SN, the plane consisting of sella and nasion; GoMe, the plane consisting of gonion and menton; U1, upper central incisor; L1, lower central incisor; E line, a line drawn from pronasale to soft tissue pogonion.

## Data Availability

All relevant data are within the manuscript.

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
