# Peer review of "Orthopedic and Nonsurgical Orthodontic Treatment of Adolescent Skeletal Class III Malocclusion Using Bone-Anchored Maxillary Protraction and Temporary Anchorage Devices: A Case Report"

_children, 2022, doi:10.3390/children9050683_

Round 1

Reviewer 1 Report

Dear authors
I believe that in this case report it is necessary to discuss in more detail the decision made in relation to the impacted tooth in the maxilla, as well as the significant deflection of the maxillary incisors.
I believe the treatment is well done, but these are two important issues that are not adequately addressed.

Sincerely Yours

Reviewer

Author Response

Reviewer 1:

I believe that in this case report it is necessary to discuss in more detail the decision made in relation to the impacted tooth in the maxilla, as well as the significant deflection of the maxillary incisors.
I believe the treatment is well done, but these are two important issues that are not adequately addressed.

Thank you very much for your comment. The relevant corrections are as follows:

  1. Case presentation:

2.1. Diagnosis and Etiology

As shown in Figure 2C, it was clearly confirmed that the radiopacity of the maxillary sinus around the right impacted tooth was increased differently from that of the left maxillary sinus. Therefore, reverse direction of the premolar towards the maxillary sinus had caused perforation of the sinus wall resulting in right sinusitis.

2.3. Treatment Alternatives

First, right maxillary sinusitis was caused by the inverted impacted tooth that penetrated the sinus wall. Although maxillary sinusitis cannot be treated directly at the orthodontic department, it was confirmed that the impacted maxillary right second premolar was the cause of right maxillary sinusitis, and that if the cause was removed, the maxillary sinusitis could be improved. Moreover, the prognosis would be poor if orthodontic traction of the tooth was performed. Therefore, tooth extraction was selected as the best treatment option.

  1. Treatment Progress

After 39 months of secondary orthodontic treatment, all spaces were closed, the anterior crossbite and lateral open bite improved, and stable occlusion was achieved (Figure 6). After treatment, most cephalometric values, except SNA, improved to within the normal range (Figure 7A, Table 1). Panoramic radiography demonstrated that the mandibular right third molar had erupted in an upright position (Figure 7B). No significant root resorption was observed in the maxillary anterior teeth, but the periodontal ligament was clearly widened. It was observed that the right maxillary sinus pneumatization was severe up to the place where the impacted tooth was in the past.

  1. Discussion

The primary second molars were present on the right side of the maxilla and mandible. Thus, in the second phase of treatment, it was important to decide whether to maintain or extract the teeth. The primary mandibular second molar was covered by a metal crown with severe root resorption, which may be considered a poor prognosis. Additionally, straight eruption of the third molar could be expected if the primary molar in the mandible was extracted and a good occlusal relationship could be established. There was almost no mobility of the maxillary primary tooth at the start of the second phase. Moreover, severe right maxillary sinus congestion was also noted (sinus pneumatization) because of the impacted second premolar. Therefore, protraction of the posterior teeth in this area would have been associated with various side effects, such as root resorption and perforation of the maxillary sinus wall during prolonged treatment. Therefore, only the mandibular primary tooth was extracted while maintaining the maxillary primary tooth.

The limitations of this case are as follows. Fortunately, no adverse events such as temporomandibular disorder or gingival recession in the mandibular anterior region occurred before and after the use of bone-anchored maxillary protraction in this patient. However, the treatment protocol that can prevent the possibility of such side effects was also performed without being prepared, so it is necessary to secure a response for this in the future. Fortunately, no significant root resorption was observed while attempting the labial inclination of the maxillary anterior teeth in the second phase, but the periodontal ligament was clearly widened, and the labial inclination was also caused by the unintentional and uncontrolled tipping movement. Therefore, to clearly confirm whether the treatment result of this case is stably maintained, it is thought that at least 1-2 years of maintenance period observation is necessary.

Reviewer 2 Report

The article “Orthopedic and Nonsurgical Orthodontic Treatment of Adolescent Skeletal Class III Malocclusion Using Bone-anchored Maxillary Protraction and Temporary Anchorage Devices: A Case Report” is very interesting.

I recommend the authors to add the limitation of the study due to a short follow-up period. Complications on the temporomandibular join and on periodontal tissue are possible. Although the anterior teeth roots were in a good condition, periodontal ligament widening is already evident.

Author Response

Reviewer 2:

The article “Orthopedic and Nonsurgical Orthodontic Treatment of Adolescent Skeletal Class III Malocclusion Using Bone-anchored Maxillary Protraction and Temporary Anchorage Devices: A Case Report” is very interesting.

I recommend the authors to add the limitation of the study due to a short follow-up period. Complications on the temporomandibular join and on periodontal tissue are possible. Although the anterior teeth roots were in a good condition, periodontal ligament widening is already evident.

Thank you very much for your comment. The relevant corrections are as follows:

  1. Discussion

The limitations of this case are as follows. Fortunately, no adverse events such as temporomandibular disorder or gingival recession in the mandibular anterior region occurred before and after the use of bone-anchored maxillary protraction in this patient. However, the treatment protocol that can prevent the possibility of such side effects was also performed without being prepared, so it is necessary to secure a response for this in the future. Fortunately, no significant root resorption was observed while attempting the labial inclination of the maxillary anterior teeth in the second phase, but the periodontal ligament was clearly widened, and the labial inclination was also caused by the unintentional and uncontrolled tipping movement. Therefore, to clearly confirm whether the treatment result of this case is stably maintained, it is thought that at least 1-2 years of maintenance period observation is necessary.

Round 2

Reviewer 1 Report

Dear Authors

In my opinion this paper is ready for publication

Congratulations 

Reviewer